# Distribution of Human Papillomavirus (HPV) Genotypes in HIV-Negative and HIV-Positive Women with Cervical Intraepithelial Lesions in the Eastern Cape Province, South Africa

**DOI:** 10.3390/v13020280

**Published:** 2021-02-11

**Authors:** Ongeziwe Taku, Zizipho Z. A. Mbulawa, Keletso Phohlo, Mirta Garcia-Jardon, Charles B. Businge, Anna-Lise Williamson

**Affiliations:** 1Division of Medical Virology, Department of Pathology, Faculty of Health Sciences, University of Cape Town, Cape Town 7925, South Africa; tkxong001@myuct.ac.za (O.T.); Zizipho.Mbulawa@nhls.ac.za (Z.Z.A.M.); phhkel001@myuct.ac.za (K.P.); 2Institute of Infectious Disease and Molecular Medicine, University of Cape Town, Cape Town 7925, South Africa; 3SAMRC Gynaecological Cancer Research Centre, University of Cape Town, Cape Town 7925, South Africa; 4Department of Laboratory Medicine and Pathology, Walter Sisulu University, Mthatha 5100, South Africa; 5National Health Laboratory Service, Nelson Mandela Academic Hospital, Mthatha 5100, South Africa; 6Department of Pathology, Walter Sisulu University and National Health Laboratory Service, Mthatha 5100, South Africa; mirta@tiscali.co.za; 7Department of Obstetrics and Gynaecology, Nelson Mandela Academic Hospital, Mthatha 5100, South Africa; cbusingae@gmail.com; 8Department of Obstetrics and Gynaecology, Faculty of Health Sciences, Walter Sisulu University, Mthatha 5100, South Africa

**Keywords:** human papillomavirus, human immunodeficiency virus, cervical intraepithelial lesions, South Africa

## Abstract

South African women have a high rate of cervical cancer cases, but there are limited data on human papillomavirus (HPV) genotypes in cervical intraepithelial neoplasia (CIN) in the Eastern Cape province, South Africa. A total of 193 cervical specimens with confirmed CIN from women aged 18 years or older, recruited from a referral hospital, were tested for HPV infection. The cervical specimens, smeared onto FTA cards, were screened for 36 HPV types using an HPV direct flow kit. HPV prevalence was 93.5% (43/46) in CIN2 and 96.6% (142/147) in CIN3. HIV-positive women had a significantly higher HPV prevalence than HIV-negative women (98.0% vs. 89.1%, *p* = 0.012). The prevalence of multiple types was significantly higher in HIV-positive than HIV-negative women (*p* = 0.034). The frequently detected genotypes were HPV35 (23.9%), HPV58 (23.9%), HPV45 (19.6%), and HPV16 (17.3%) in CIN2 cases, while in CIN3, HPV35 (22.5%), HPV16 (21.8%), HPV33 (15.6%), and HPV58 (14.3%) were the most common identified HPV types, independent of HIV status. The prevalence of HPV types targeted by the nonavalent HPV vaccine was 60.9% and 68.7% among women with CIN2 and CIN3, respectively, indicating that vaccination would have an impact both in HIV-negative and HIV-positive South African women, although it will not provide full protection in preventing HPV infection and cervical cancer lesions.

## 1. Introduction

Human immunodeficiency virus (HIV) infection is highly predominant in Africa. Globally, approximately 25.7 million people live with HIV infection, of which 80% (20.7 million) reside in Eastern and Southern Africa, as reported in 2019 [1,2]. Of the African countries, South Africa has the largest population affected by HIV infection, with 7.5 million people living with HIV and 200,000 new infections reported in 2019 [3]. HIV prevalence is estimated to be 19.0% among women aged 15–49 years [2,3]. Among the nine provinces in South Africa, the prevalence of HIV infection for adult women aged 15–49 years ranges from 12.6% to 27.0%, with high rates observed in Kwazulu-Natal province (27.0%), Free State province (25.5%), and Eastern Cape province (25.2%) [4]. 

An interaction between HIV infection and specific cancers has been established. Cervical cancer is one of the three cancers established as AIDS-defining cancers [5]. South Africa ranked as the country with the fourth highest number of cervical cancer cases among HIV-positive women (63.4%) in 2018 [6]. The incidence rate of cervical cancer was estimated to be 506 per 10,000 person-years among HIV-positive South African women in 2017 [7]. Cervical cancer arises from cervical intraepithelial neoplastic lesions (CIN) stages 1–3 and is causally associated with genital human papillomavirus (HPV) [8,9]. HIV-positive women have an increased burden of genital HPV acquisition, high-risk (HR) HPV persistent infection, multiple infections of HR-HPV, and precancerous lesions compared to HIV-negative women [10,11]. Studies suggest that this results from immune suppression and low CD4 cell count [10,12].

Persistent infection with HPV16, HPV18, and other HR-HPV genotypes is the most significant risk factor for developing cervical lesions and cervical cancer [13]. HPV16/18 are essential HR-HPV types that significantly contribute to cervical cancer disease progression [14]. There are different strategies implemented to prevent preinvasive lesions and cervical cancer, mainly through HPV vaccination and cervical cancer screening. The type-specific HPV vaccines, namely, bivalent (HPV16/18), quadrivalent (HPV16/18/6/11), and nonavalent (HPV16/18/6/11/31/33/45/52/58), have been introduced in more than forty countries, both developing and developed [15,16,17]. These vaccines are offered to adolescent and young women aged 9–26 years [15,16]. In South Africa, a national school-based vaccination campaign for the bivalent HPV vaccine was implemented to target public school girls aged nine years in grade 4. An uptake of the bivalent HPV vaccine ranging from 87% to 92% was positively attained among South African girls [18,19]. However, cervical cancer screening among older women is still necessary, as they are beyond the targeted age for receiving the HPV vaccine and more likely to have been infected with HPV.

Molecular HPV DNA testing has been implemented as the alternative to non-molecular testing for cervical cancer, particularly cytology testing [20]. HPV testing is utilised in various strategies for screening, such as triage, co-testing, or HPV testing alone [21], and has high sensitivity but low specificity for the detection of CIN2/3 [22,23].

There is limited information on the epidemiology of HPV types in women with preinvasive cervical lesions from the Eastern Cape province. It is essential to investigate the prevalence of HPV genotypes and their distribution among women with different immune statuses and confirmed CIN histology results in this region. Therefore, these data will help achieve better understanding of the HR-HPV types involved in cervical lesions and cervical cancer cases. Furthermore, this information will contribute to discussions about implementing strategies for cervical screening and monitoring HPV types that are not present in the current vaccines to reduce cervical cancer disease in this population. Our study aims at investigating the distribution of HPV genotypes among HIV-positive and HIV-negative women with cervical intraepithelial lesions from Eastern Cape, South Africa.

## 2. Material and Methods 

### 2.1. Study Population

The study obtained ethical approval from the Human Ethics Committees of the University of Cape Town (UCT, HREC reference 615/2017), Walter Sisulu University (016/2017), and Eastern Cape Department of Health (EC reference 2017RPO_484). The recruitment procedure for this study was reported previously [24]. Briefly, between September 2017 and March 2019, cervical specimens were collected among women referred to the Nelson Mandela Academic Hospital Gynaecology Outpatient Clinic located in the OR Tambo municipality area in the Eastern Cape province, South Africa. A total of 193 women were recruited, aged ≥18 years, with atypical squamous cells of undetermined significance (ASCUS), atypical squamous cells cannot exclude high-grade lesions (ASC-H), atypical glandular cells, not otherwise specified (AGC-NOS), low-grade squamous intraepithelial lesions (LSIL), and high-grade squamous intraepithelial lesions (HSIL). The cervical specimens were collected by a study nurse using a Viba-brush (Rovers Medical Devices B.V., 5347 KV Oss, Netherlands, smeared onto FTA cards (GE Healthcare, Amersham place little Chalfont, Buckinghamshire HP7 9NA, UK), and shipped at room temperature to UCT. The cervical biopsy was collected for histopathology and was performed by the National Health Laboratory Service. Based on the histopathology results, 46 women with CIN2 and 147 with CIN3 were included in this study. All eligible women provided signed consent forms.

### 2.2. Detection of HPV Genotypes

DNA elution of cervical specimens from FTA cards was done following the procedure previously described [24,25]. Four microlitres of extracted DNA was used for HPV testing. Detection of HPV genotypes was performed using an HPV direct flow chip kit on a Hybrispot machine (Master Diagnostica, Granada, Spain) following the manufacturer’s procedure. The HPV direct flow chip protocol is a PCR-based method based on the amplification of a viral DNA fragment, followed by hybridisation onto a membrane chip using the amplified PCR products. The chip membrane contains DNA control, hybridisation control, PCR control, and probes for genotype-specific HPV detection. The assay detects 36 HPV genotypes (low-risk HPV: 6, 11, 40, 42, 43, 44, 54, 55, 61, 62, 67, 69, 70, 71, 72, 81, 84, and 89 (C6108) and high-risk HPV:16, 18, 26, 31, 33, 35, 39, 45, 51, 52, 53, 56, 58, 59, 66, 68, 73 and 82). Each chip membrane’s results were captured by a camera and analysed automatically using HybriSoft software (Master Diagnostica, Granada, Spain) [26].

### 2.3. Data Analysis

All data and statistical analyses were done using GraphPad Prism version 6 (GraphPad Software, La Jolla, CA, USA). The chi-squared test was used to determine a statistical difference between HPV infection and variables. A variable was considered significant if the *p*-value was < 0.05. 

## 3. Results

### 3.1. Description of Study Participants

The median age of women was 40 (IQR: 33–48) years. A high number of women were HIV-positive (76.2%) and never smoked (93.3%), and half of the women had their first sexual experience at the age of 16–18 years (53.9%) (Table 1). Women were more likely to have ≥3-lifetime sexual partners (62.7%), with a high proportion having high-grade squamous lesions on cytology testing (75.1%) (Table 1). 

### 3.2. HPV Prevalence According to HIV Status

Of the 193 women screened, 93.5% (43/46) with CIN2 and 96.6% with CIN3 had an HPV infection. HIV-positive women had a significantly higher prevalence of any HPV infection compared to HIV-negative women (98.0% vs. 89.1%, *p* = 0.012) (Table 2). HIV-negative women were almost 2-times more likely to have only a single HPV infection compared to HIV-positive women, although there was no statistical significance (OR: 1.45; CI: 0.735–2.867, *p* = 0.282, Table 2). However, HIV-positive women had a significantly higher HPV prevalence compared to HIV-negative women (65.3% vs. 47.8%, *p* = 0.034, Table 2). For multiple infections, the median of HPV types was 2 (range: 2–11). When stratified by HIV status, there was no significant distinction between HIV-negative and HPV-positive women (Figure 1).

### 3.3. HPV Distribution According to Cervical Intraepithelial Lesions and HIV Status

Women with CIN2 were more likely to be infected with two HPV types (Figure 2A). The most frequently detected HPV types were HPV35 (23.9%), HPV58 (23.9%), HPV45 (19.6%), and HPV16 (17.3%) (Table 3). Among single HPV infections, HPV35 and HPV16 were more frequent in HIV-negative women, while HPV16 and HPV52 were detected in HIV-positive women (Figure 3). For multiple HPV infections, HPV35, HPV58, and HPV45 were common in HIV-positive women, whereas HPV35, HPV58, and HPV16 were frequently detected among HIV-negative women (Figure 3).

Women with CIN3 were more likely to be infected with two HPV types (Figure 2B). Their commonly identified HPV types were HPV35 (22.5%), HPV16 (21.8%), HPV33 (15.6%), and HPV58 (14.3%) (Table 3). In single infections, HPV16, HPV35, and HPV33 were mostly observed in HIV-negative and HIV-positive women (Figure 4). However, the most detected HPV types in multiple infections were HPV16, HPV35, and HPV66 in HIV-negative, whereas HPV16, HPV35, and HPV45 were observed among HIV-positive women (Figure 4).

### 3.4. HPV Prevalence According to Vaccine HPV Types

The prevalence of bivalent vaccine HPV types (HPV16/18) increased from 20.0% of CIN2/HIV-negative to 25.0% of CIN2/HIV-positive and from 27.8% of CIN3/HIV-negative to 29.7% of CIN3/HIV-positive, with no statistical significance (*p* = 1.000 and *p* = 0.823, respectively) (Figure 5). For quadrivalent HPV types (HPV6/11/16/18), the positivity increased from 20.0% for CIN2/HIV-negative to 30.5% for CIN2/HIV-positive and from 33.3% for CIN3/HIV-negative to 36.0% for CIN3/HIV-positive (*p* = 0.700 and *p* = 0.768, respectively) (Figure 5). CIN2/HIV-positive women had a higher prevalence of nonavalent HPV types than CIN2/HPV-negative, with no statistical significance (66.7% vs. 40.0%, *p* = 0.157). However, CIN3/HIV-negative women had a similar prevalence of nonavalent HPV types (HPV6/11/16/18/31/33/45/52/58) to CIN3/HIV-positive women (67.7% vs. 69.0%, *p* = 0.761) (Figure 5).

## 4. Discussion

This study investigated the prevalence and distribution of HPV types among HIV-positive and HIV-negative women with high-grade precancerous cervical lesions. A high number of women in this study were HIV-positive (76.2%). A significantly high overall prevalence of any HPV infection (98% vs. 89%) and multiple infections (65% vs. 49%) was observed among HIV-positive compared to HIV-negative women with cervical intraepithelial lesions. This higher prevalence of multiple types of HPV infection among women with HIV infection agrees with other cohorts of women with high-grade lesions from Botswana and South Africa [27,28,29]. However, in a South African study by Van Aardt and colleagues (2016), the rate of multiple HPV infections among HIV-positive and HIV-negative women (81.3% vs. 64.4%) with confirmed CIN2/3 was higher compared to our study [29]. This difference could be attributed to the various assays used for HPV testing and the different study populations. 

A high prevalence of HPV in women with CIN2 (93.5%) and CIN3 (96.6%) was observed in this study, which is expected as these women had abnormal cytology from the referral clinic. Similarly, a high HPV prevalence was reported in a global meta-analysis study, whereby HPV prevalence ranged from 86% to 93% in women with high-grade lesions (CIN2/3) [30]. In the present cohort, HPV35 was the most predominant HPV type among participants with CIN2 lesions, while in those with CIN3 lesions, HPV16 and HPV35 were the most frequently detected HPV genotypes, either as a single HPV infection or multiple infections, regardless of HIV status. This observation was similar to other studies from South Africa and Kenya, where HPV16 and 35 were the most common genotypes in HIV-positive or HIV-negative women with high-grade squamous intraepithelial lesions or CIN2/3 [28,31]. However, a study among sex workers from Kenya showed that HPV52 was the most prevalent HPV type and more likely to be present as a single infection in women with severe lesions (HSIL/SCC) [32]. Furthermore, a recent cross-sectional study by Dovey (2018) among women from four developed countries (Iceland, Norway, Sweden, and Denmark) reported a different distribution of HPV types, with HPV16, 31, and 52 present in CIN2 cases and HPV 16, 31, and 33 detected in CIN3 cases [33]. The high occurrence of HPV35 in this population and other African studies of women with invasive cervical cancer suggests an interaction between HPV35 and cervical carcinogenesis [34,35,36]. Therefore, preventative strategies are needed, as the HPV35 genotype is present in up to 10% of sub-Saharan African women with invasive cervical cancer [36,37,38,39] and not present in the current HPV vaccines.

Previously, cervical histological lesions were associated with one HPV type [40]. However, in the present cohort, most high-risk HPV types and probable high-risk HPV types occurred as multiple HPV infections both in CIN2 and CIN3 cases. Multiple infections are reported as the risk factor of persistent infection and associated with high-grade CIN2/3 cases compared to a single infection [41]. Women with multiple HPV types have been found to have larger cervical lesions and are associated with poor responses to cervical cancer treatment [42,43]. Kaliff et al. (2018) reported a significantly high recurrence rate of cervical cancer among women with multiple HPV infections compared to a single HPV infection (44.0% vs. 24.0%) and a low cancer survival rate [42]. Furthermore, the high prevalence of multiple HPV infections in the present study could be because HPV testing was performed on cervical cells instead of biopsy specimens. Therefore, it is not possible to determine which HPV types caused the lesion and which infected other parts of the cervix. HPV testing on biopsies eliminates the detection of multiple HPV infections, and multiple HPV infections are observed to be significantly lower in biopsies compared to exfoliated cells from invasive cervical cancers [39,44].

Interestingly, in our study, HPV16 was not the predominant HPV type, as it ranked fourth in CIN2 and second in CIN3 cases in the present cohort. A low prevalence of HPV16 has been observed in other sub-Saharan studies, while a high prevalence was observed in European studies [45,46,47]. These findings could be explained by the population being sampled from different geographical areas, host genetic difference/host immunogenic factors, and the biological interplay between HPV types [48]. It is important to do a study based on cervical cancer biopsies to determine which HPV types are causally involved in cervical cancer in this community. The distribution of HPV types in this study are concerning, as there are many types that are not in the available vaccines.

Persistent HR-HPV infection is regarded as a significant factor in the development of cervical cancer lesions. However, in the current study, 4.2% (8/193) of women with CIN2/3 were negative for any HPV type, and seven of these samples were also negative when typed with *hpVIR* real-time PCR [24]. One specimen was positive on *hpVIR* real-time PCR for HR-HPV infection (HPV59) but had a low HPV copy number (11.3 copies) and viral titre (0.052). The observed negative results of HPV infection in women with high-grade lesions may suggest that it could result from sample storage, inadequate sampling, or low viral load. Alternatively, there may be novel HPV types causing the cancers that are not detected by the test used.

The currently available HPV vaccines are estimated to prevent 70–90% of cervical cancer cases [49]. The nonavalent HPV vaccine has been highly effective in preventing HPV infection and cancer diseases, with efficacies ranging between 90% and 100% [15,50,51,52]. A study reported by Garland and colleagues (2018) in Asian women showed that nonavalent significantly decreased the risk of persistent infection, abnormal cytology, and diseases caused by specific HPV types targeted by this vaccine [50]. In the present study, HPV vaccines could protect 20–69.0% of CIN2/3 cases in women with or without HIV infection. Therefore, the high prevalence of HPV types targeted by the nonavalent HPV vaccine suggests that introducing this HPV vaccine would be beneficial, as most precancerous lesions could have been prevented. However, the predominant HPV genotype (HPV35) in this population, which accounts for 24% in CIN2 and 23.0% in CIN3, is not covered by the nonavalent HPV vaccine. 

Vaccines have been found to provide cross-protection against specific vaccine HPV types as well as some types that are not present in the vaccine. Numerous trial studies have reported that the bivalent vaccine offers a wider extent of cross-protection against nonvalent specific types (31/33/45/52/58), with less extensive cross-protection by Gardasil-9 [53,54,55]. Studies have reported that bivalent showed substantial cross-protection against HPV31/33/45 but less so against HPV35 and HPV58 after seven to eight years postvaccination [53,56,57]. A study by Brown and colleagues (2009) among younger women aged 16–25 years vaccinated with the quadrivalent HPV vaccine showed a reduction of high-grade lesions (32.5%) related to ten non-vaccine HPV types (31/33/35/39/45/52/52/56/58/59) known to cause cervical cancer after 3.6 years of follow-up [58]. Therefore, this suggests that cross-protection might have played a role and that the benefits of vaccination could include protection from clinically relevant HPV types not included in the vaccines [58]. The cross-protection is related to the phylogenetic distance between the HPV types, as they are all closely related to vaccine types and found in the alpha-9 group [56]. However, since cross-protection against HPV35 is observed to be less efficient compared to other HPV types, the addition of HPV35 to the next-generation HPV vaccine would improve the effectiveness of the HPV vaccine, especially in Africa. 

## 5. Conclusions 

We observed a significantly higher prevalence of HPV and multiple HPV infections in HIV-positive compared to HIV-negative women with cervical intraepithelial lesions. The distribution of HPV genotypes was similar between CIN2 and CIN3 cases independently of HIV status. The HPV nonavalent vaccine would have an impact on South African women, although it will not provide full protection in preventing HPV infection and cervical cancer lesions. Therefore, the high prevalence of the non-vaccine type (HPV35) underscores the need to incorporate this HPV type into the next HPV vaccines. This study also highlights the importance of introducing cervical cancer screening strategies to monitor non-vaccine HPV types. 

## Figures and Tables

**Figure 1 viruses-13-00280-f001:**
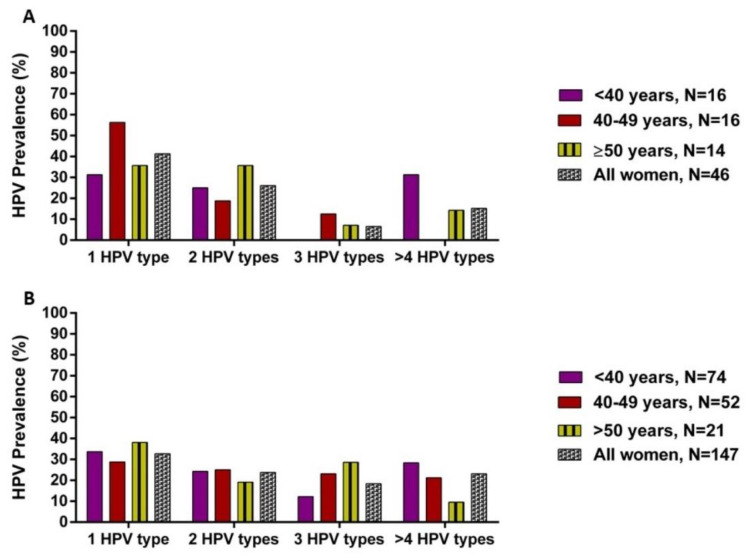
Distribution of single and multiple HPV types according to age in HIV-negative (**A**) and HIV-positive (**B**) women with cervical intraepithelial lesions.

**Figure 2 viruses-13-00280-f002:**
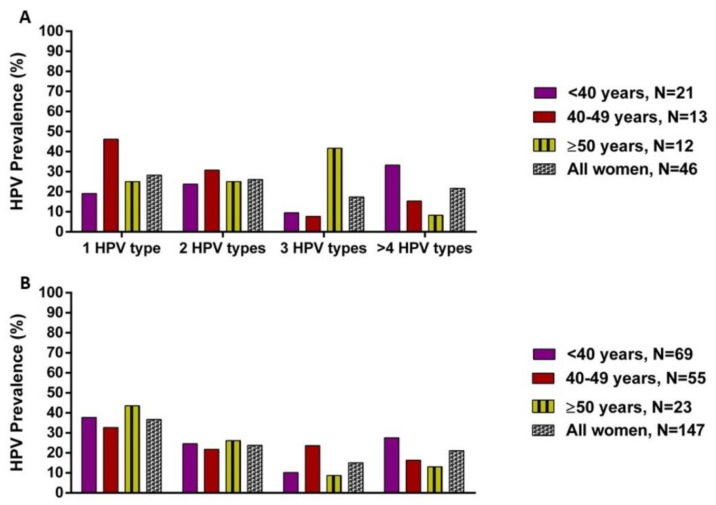
Distribution of single and multiple HPV types according to age in women with CIN2 (**A**) and CIN3 (**B**).

**Figure 3 viruses-13-00280-f003:**
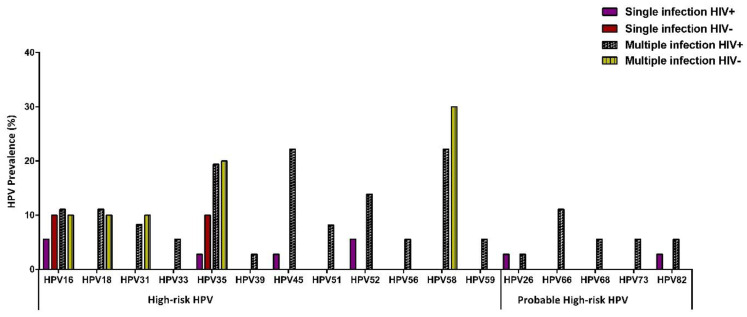
Distribution of HPV types among women with CIN2 in single and multiple infections according to HIV status.

**Figure 4 viruses-13-00280-f004:**
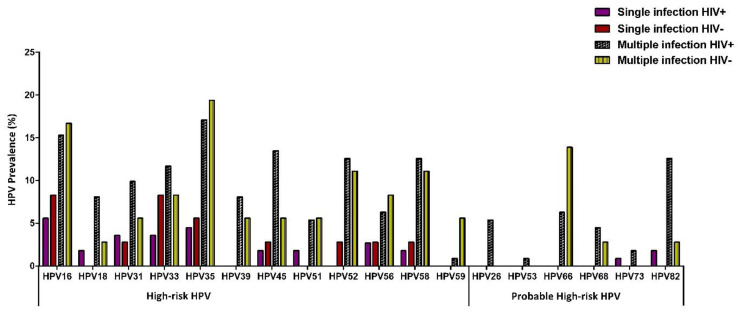
Distribution of HPV types among women with CIN3 in single and multiple infections according to HIV status.

**Figure 5 viruses-13-00280-f005:**
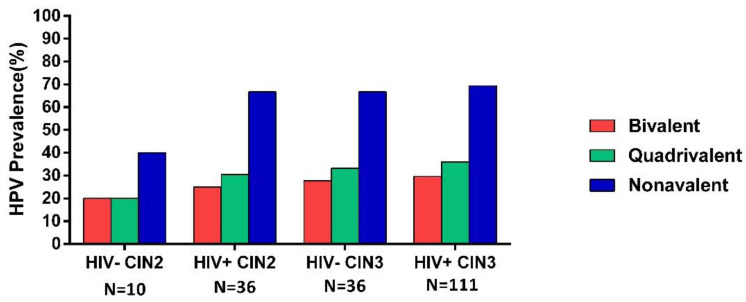
Prevalence of HPV vaccine types according to CIN2/3 and HIV status.

**Table 1 viruses-13-00280-t001:** Demographic and behavioural characteristics of study participants.

Variables	% (n/N)
**Age in years: Median (IQR)**	40 (33–48)
**HIV Status**	
No	23.8% (46/193)
Yes	76.2% (147/193)
**Age categories**	
18–29 years	11.4% (22/193)
30–39 years	35.2% (68/193)
40–49 years	35.2% (68/193)
≥50 years	18.1% (35/193)
**Highest level of education attained**	
Never/primary	29.5% (57/193)
High school/university	70.5% (136/193)
**Household income**	
<$139.36	75.1% (145/193)
≥$139.36	22.8% (44/193)
**Smoking status**	
Never	93.3% (180/193)
Former/current smoker	6.2% (12/193)
**Age at first sexual experience**	
<16 years	21.2% (41/193)
16–18 years	53.9% (104/193)
≥18 years	24.9% (48/193)
**Lifetime sexual partners**	
1	15.0% (29/193)
2	21.8% (42/193)
≥3	62.7% (121/193)
**Cytology**	
ASCUS/ASCU-H/AGC-NOS	13.5% (26/193)
LSIL	9.3% (18/193)
HSIL	75.1% (145/193)

ASCUS: atypical squamous cells of undetermined significance; ASC-H: atypical squamous cells cannot exclude high-grade lesions; AGC-NOS: atypical glandular cells, not otherwise specified; LSIL: low-grade squamous intraepithelial lesions; HSIL: high-grade squamous intraepithelial lesions.

**Table 2 viruses-13-00280-t002:** Prevalence of HPV infection according to HIV status.

Variables	HIV-Negative, N = 46	HIV-Positive, N = 147	OR (95%CI)	*p*-Value
Any type	89.1% (41/46)	98.0% (144/147)	0.17 (0.039–0.745)	0.012
Single infection	41.3% (19/46)	32.7% (48/147)	1.45 (0.735–2.867)	0.282
Multiple infection	47.8% (22/46)	65.3% (96/147)	0.49 (0.249–0.953)	0.034
HR-HPV types	82.6% (38/46)	87.1% (128/147)	0.71 (0.286–1.738)	0.446
Probable HR-HPV types	17.4% (8/46)	30.6% (45/147)	0.60 (0.260–1.394)	0.233
LR-HPV	39.1% (18/46)	44.2% (65/147)	0.81 (0.413–1.594)	0.543

HR-HPV: high-risk human papillomavirus; LR-HPV: low-risk human papillomavirus; OR: odds ratio; CI: confidence intervals.

**Table 3 viruses-13-00280-t003:** Distribution of HPV genotypes among women with CIN2 and CIN3 from a referral hospital.

HPV Types	All Women % (n/N)	CIN2 % (n/N)	CIN3 % (n/N)
16	**20.7% (40/193)**	**17.4% (8/46)**	**21.8% (32/147)**
18	8.8% (17/193)	10.9% (5/46)	8.8% (12/147)
31	11.4% (22/193)	8.7% (4/46)	12.2% (18/147)
33	13.0% (25/193)	4.4% (2/46)	**15.6% (23/147)**
35	**22.8% (44/193)**	**23.9% (11/46)**	**22.5% (33/147)**
39	6.2% (12/193)	2.2% (1/46)	7.5% (11/147)
45	**15.0% (29/193)**	**19.6% (9/46)**	13.6% (20/147)
51	6.7% (13/193)	6.5% (3/46)	6.8% (10/147)
52	13.5% (26/193)	15.2% (7/46)	12.9% (19/147)
56	8.3% (16/193)	4.4% (2/46)	9.5% (14/147)
58	**16.6% (32/193)**	23.9% (11/46)	**14.3% (21/147)**
59	2.6% (5/193)	4.4% (2/46)	2.0% (3/147)
26	4.1% (8/193)	4.4% (2/46)	4.1% (6/147)
53	0.5% (1/193)	0.0% (0/46)	0.7% (1/147)
66	8.3% (16/193)	8.7% (4/46)	8.2% (12/147)
68	4.1% (8/193)	4.4% (2/46)	4.1% (6/147)
73	2.6% (5/193)	4.4% (2/46)	2.0% (3/147)
82	10.4% (20/193)	6.5% (3/46)	11.6% (17/147)
6	6.2% (12/193)	6.5% (3/46)	6.1% (9/147)
11	4.7% (9/193)	0.0% (0/46)	6.1% (9/147)
40	3.6% (7/193)	2.2% (1/46)	4.1% (6/147)
42	7.3% (14/193)	8.7% (4/46)	6.8% (10/147)
43	2.1% (4/193)	0.0% (0/46)	2.7% (4/147)
44/55	10.9% (21/193)	**17.4% (8/46)**	8.8% (13/147)
54	3.6% (7/193)	6.5% (3/46)	2.7% (4/147)
61	0.5% (1/193)	0.0% (0/46)	1.4% (1/147)
62/81	15.0% (29/193)	**19.6% (9/46)**	13.6% (20/147)
70	4.1% (8/193)	6.5% (3/46)	3.4% (5/147)
71	4.7% (9/193)	2.2% (1/46)	5.4% (8/147)
72	3.6% (7/193)	6.5% (3/46)	2.7% (4/147)

HPV: human papillomavirus; CIN: cervical intraepithelial neoplasia. Bold indicates the most dominant HPV types.

## Data Availability

The data analysed in this study is available upon request from the corresponding author.

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
