# Peer review of "Distribution of Human Papillomavirus (HPV) Genotypes in HIV-Negative and HIV-Positive Women with Cervical Intraepithelial Lesions in the Eastern Cape Province, South Africa"

_viruses, 2021, doi:10.3390/v13020280_

Round 1
Reviewer 1 Report
This paper reports on HPV-detection in 193 CIN2/3 cases divided by HIV-status from one hospital in South Africa. I think that the manuscript brings the interesting observation that HPV35 is a prevalent type in CIN2/3 cases both with and without HIV. However, this interesting observation is almost impossible to find in this extensive reporting. The paper is far too long.
I recommend the authors to re-write their findings as a short communication.
Author Response
Response to Reviewer 1 Comments
Point 1: This paper reports on HPV-detection in 193 CIN2/3 cases divided by HIV-status from one hospital in South Africa. I think that the manuscript brings the interesting observation that HPV35 is a prevalent type in CIN2/3 cases both with and without HIV. However, this interesting observation is almost impossible to find in this extensive reporting. The paper is far too long.
Response 1: Some paragraphs and figures on the manuscript were taken out as they have limited value with such low numbers (This will not impact on the paper) and to shorten the manuscripts as advised by the reviewer.
Line 49-51 “Of the 569 847 new cervical cancer cases that occurred globally, 33000 new cases were found in HIV-positive women, with 63.8% of these cases occurring in Southern Africa in 2018 [6].”
Line 53-55 “In a study done by Rohner and colleagues (2017) among South African women with HIV infection, the overall incidence rate of cervical cancer was estimated to be 506 per 100 000 person-years [7]”
Line 56-57 “It has been identified that the development of cervical cancer lesions is”
Line 65-69 “Approximately 3.2% of South African women in the general population will get infected with HPV16/18 infection at some point in their lives [15]. An estimated 64.2% of invasive cervical cancer cases in South Africa are attributed to HPV16/18 [15]. The prevalence of these HR-HPV genotypes is observed to increase with the severity of preinvasive stages [16].”
Line 81-82 “Furthermore, cervical cancer screening is vital to monitor nonvaccine HPV types as they are associated with cervical cancer lesions/cytology abnormalities”
Line 155-156 “women were more likely to be infected with 2 HPV types (23.8%) and 4 HPV types (23.1%)”
Line 164-166 “Of these women, HIV-positive women age 40-49 years were having higher HPV prevalence (30.8%) (Figure 2&3).”
Line 173-176 “According to HIV status, HIV-negative women were more likely to have 2 HPV types, with a high prevalence observed among women age 50 years or more (36.4%) (Figure 4). Similarly, HIV-positive women were more likely to have two types, and the prevalence decreased with increasing age (Figure 4).”
Figure 3. Distribution of single and multiple HPV types according to age in HIV-negative/CIN2 (a) and HIV-positive/CIN2 women (b).
Figure 4. Distribution of single and multiple HPV types according to age in HIV-negative/CIN3 (a) and HIV-positive/CIN3 women (a).
Line 230-239 “HIV-positive status has a negative impact on HPV prevalence and has been previously reported as the independent risk factor of HPV infection [33, 34]. Similarly, HIV-positive women with high HIV viral of more than 100 000 copies/ml and CD4 cell count of more than 500 cell/mm3 were more likely to acquire multiple HPV infections [35]. Additionally, it has been reported that the immunologic mechanism plays a significant role in increasing HPV infection among HIV-positive women [36, 37]. Studies reporting a high prevalence of multiple infections among HIV-positive women suggest that it could result from being unable to clear the infection or reactivation of latent HPV infection due to immunosuppression [35, 38]. Therefore, these findings confirm that this group of women is a priority, and screening should be performed in all infection stages.”
Line 271-273 “Similarly, a study done in Kenya showed that the prevalence of multiple infections ranged 2-4 times higher in cervical cells than biopsies of HIV-positive women [51]. Also, the prevalence of”

Reviewer 2 Report
I’m not sure that this work breaks terribly new ground, except for a specific geographical area, but that may be valuable on its own.
It seems odd to me that the percentage given for the smaller population, 93.5% (43/46), has more significant figures than the data. Similar checks should be carried out for the other percentages quoted (three significant figures are valid with a data set in the hundreds, not in the tens).
A full stop is needed at the end of line 114.
Line 117, ul should be uL.
Line 149, for emphasis and clarity, I think this should say ,”…ative women were almost 2-times more likely to have ONLY a single HPV infection compared to…” The CAPS were put in for emphasis. This is assuming I understand the point.
153-156 I don’t understand the need to make the following distinctions. They are not very significant, they vary with age, and probably do not mean much for these sample sizes:
“153 When stratified by HIV sta-
154 tus, HIV-negative women were more likely to be infected with 2 HPV types (26.1%). HIV
155 positive women were more likely to be infected with 2 HPV types (23.8%) and 4 HPV
156 types (23.1%, Figure 1).”
I don’t understand the labeling of Fig 3. For A, “All women” is labeled N= 10, but it clearly isn’t equal to ten for one or two HPV types. For B, “All women” is labeled N= 36, but this is not the case for the individual parts of the figure. I’m not sure what is going on. The same problems occur for Fig. 4. A value needs to agree with the scale of the figure in all cases.
I understand that the authors want to analyze the data in detail, but I think they might be obscuring some aspects of the data by not reporting a simpler analysis of the prevalence of HPV types for HIV positive and negative patients in the absence of any other data parsing. In other words, what are the simple numbers for HPV16, 18, etc. prevalence, regardless of single or multiple viral types being present? Would HPV 16 become the most prevalent type when looked at this way? We should know the answer to this simple question for this and the other HPV types studied.
231-233 The font changes in this section and there are some typos: ml should be mL and mm3 should have the 3 as a superscript:
“ Similarly, HIV
232 positive women with high HIV viral of more than 100 000 copies/ml and CD4 cell count of more
233 than 500 cell/mm3 were more likely to acquire multiple HPV infections [35]. Additionally”
There should be some comment about possible cross-reactivity between HPV35 and the nonavalent vaccine, and what is known about that. Once the problems with the confusing figures are sorted out and the other comments are addressed, this should be ready for publication.
Author Response
Response to Reviewer 2 Comments
Point 1: A full stop is needed at the end of line 114.
Response 1: A full stop was added as advised by the reviewer. Line106 “All eligible women provided signed consent forms.”
Point 2: Line 117, ul should be uL.
Response 2: The units were corrected as advised by the reviewer. Line109-112 “Four µL of extracted DNA was used for HPV testing. Detection of HPV genotypes was performed using an HPV direct flow chip kit on a Hybrispot machine (Master Diagnostica, Granada, Spain) following the manufacture’s procedure.”
Point 3: Line 149, for emphasis and clarity, I think this should say ,”…ative women were almost 2-times more likely to have ONLY a single HPV infection compared to…” The CAPS were put in for emphasis. This is assuming I understand the point.
Response 3: The word “only” was added as advised by the reviewer. Line140-143 “HIV-negative women were almost 2-times more likely to have only a single HPV infection compared to HIV-positive women, although there was no statistical significance (OR: 1.45; CI: 0.735-2.867, P=0.282, Table 2).”
Point 4: 153-156 I don’t understand the need to make the following distinctions. They are not very significant, they vary with age, and probably do not mean much for these sample sizes:
Response 4: A sentence was corrected as advised by the reviewer. Line145-147 “When stratified by HIV status, there was no significant distinction between HIV-negative and HPV-positive women (Figure 1).”
Point 5: I don’t understand the labelling of Fig 3. For A, “All women” is labelled N= 10, but it clearly isn’t equal to ten for one or two HPV types. For B, “All women” is labelled N= 36, but this is not the case for the individual parts of the figure. I’m not sure what is going on. The same problems occur for Fig. 4. A value needs to agree with the scale of the figure in all cases.
Response 5: The figures were taken out as they are meaningless with such low numbers and this will not have an impact on the manuscript.
Point 6: I understand that the authors want to analyse the data in detail, but I think they might be obscuring some aspects of the data by not reporting a simpler analysis of the prevalence of HPV types for HIV positive and negative patients in the absence of any other data parsing. In other words, what are the simple numbers for HPV16, 18, etc. prevalence, regardless of single or multiple viral types being present? Would HPV 16 become the most prevalent type when looked at this way? We should know the answer to this simple question for this and the other HPV types studied.
Response 6: A column was added on table1 to show the overall prevalence for each HPV type as advised by the reviewer.
Point 7: 231-233 The font changes in this section and there are some typos: ml should be mL and mm3 should have the 3 as a superscript:“ Similarly, HIV
232 positive women with high HIV viral of more than 100 000 copies/ml and CD4 cell count of more
233 than 500 cell/mm3 were more likely to acquire multiple HPV infections [35]. Additionally”
Response 7: Line 231-233 was removed from the manuscript to shorten the manuscript as advised by the reviewer
Point 8: There should be some comment about possible cross-reactivity between HPV35 and the nonavalent vaccine, and what is known about that. Once the problems with the confusing figures are sorted out and the other comments are addressed, this should be ready for publication.
Response 8: A paragraph was added to address the reviewer’s comment. Line277-293 “Vaccines have been found to provide cross-protection against specific vaccine HPV types as well as some types that are not present in the vaccine. Numerous trial studies re-ported bivalent vaccine to offer wider extent of cross protection against nonvalent specific types (31/33/45/52/58) with less extent of Gardasil-9 [53-55]. Studies reported that bivalent showed a substantial cross-protection against HPV31/33/45 but less against HPV35 and HPV58 after seven to eight years post vaccination [53, 56, 57]. A study done by Brown and colleagues (2009) among younger women aged 16-25 years vaccinated with quadrivalent HPV vaccine showed a reduction of high-grade lesions (32.5%) related to ten non-vaccine HPV types (31/33/35/39/45/52/52/56/58/59), known to cause cervical cancer 3.6 years of follow-up [58]. Therefore, suggesting that cross protection might have played a role and that the benefits of vaccination could include protection from clinically relevant HPV types not included in the vaccines [58]. The cross protection is related to the phylogenetic distance between the HPV types as they are all closely related to vaccine types and found in the alpha-9 group [56]. However, since cross protection against HPV35 is observed to be less efficient compared to other HPV types the addition of HPV35 into the next generation HPV vaccine would improve the effectiveness of the HPV vaccine especially in Africa.
Reviewer 3 Report
This is a single-site cross-sectional study that explored the distribution of human papillomavirus (HPV) genotypes and their association with HIV infection in women with cervical intraepithelial lesions (CIN) in the Eastern Cape province, South Africa. The authors analyzed 193 confirmed CIN samples and screened 36 HPV genotypes. The results suggested a high prevalence of HPV in CIN and an association between HPV and HIV infection may exist. There's a difference in the distribution of HPV genotypes between CIN2 and CIN3.
Generally, the whole manuscript is well prepared in plain English. The logic is easy to follow-up. There are some point s for the authors:
- As a single-center study, the authors should recognize there should be a limit in generalization. The authors are suggested to compare their findings to national statistics for better understanding.
- The estimation of the effectiveness of HPV vaccine should be more conservative. Because of the high prevalence of HPV infection in these samples, the link between HPV vaccination, HPV infection, and CIN is weak. "vaccination would reduce the risk of precancerous
lesions both in HIV-negative and HIV-positive South African women" is not supported by the current result.
Author Response
Response to Reviewer 3 Comments
Point 1: As a single-center study, the authors should recognize there should be a limit in generalization. The authors are suggested to compare their findings to national statistics for better understanding.
Response 1: A sentence was added to address the reviewer’s comment Line A high prevalence of HPV in women with CIN2 (93.5%) and CIN3 (96.6%) was observed in this study, which is expected as these women had abnormal cytology from the referral clinic. Similarly, the high HPV prevalence was reported in global meta-analysis study whereby HPV prevalence ranged from 86-93% in women with high grade lesions (CIN2/3) [30].
Point 2: The estimation of the effectiveness of HPV vaccine should be more conservative. Because of the high prevalence of HPV infection in these samples, the link between HPV vaccination, HPV infection, and CIN is weak. "vaccination would reduce the risk of precancerous lesions both in HIV-negative and HIV-positive South African women" is not supported by the current result.
Response 2: A sentence was added to address the reviewer’s comment. Line31-35 “The prevalence of HPV types targeted by nonavalent HPV vaccine was 60.9% and 68.7% among women with CIN2 and CIN3, respectively indicating that vaccination would have an impact both in HIV-negative and HIV-positive South African women, although it will not provide full protection in preventing HPV infection and cervical cancer lesions.”